# 4-Amino-TEMPO-Immobilized Polymer Monolith: Preparations, and Recycling Performance of Catalyst for Alcohol Oxidation

**DOI:** 10.3390/polym14235123

**Published:** 2022-11-24

**Authors:** Tomoki Imoto, Hikaru Matsumoto, Seiya Nonaka, Keita Shichijo, Masanori Nagao, Hisashi Shimakoshi, Yu Hoshino, Yoshiko Miura

**Affiliations:** 1Department of Chemical Engineering, Graduate School of Engineering, Kyushu University, 744 Motooka, Nishi-Ku, Fukuoka 819-0395, Japan; 2Department of Applied Chemistry, Graduate School of Engineering, Kyushu University, 744 Motooka, Nishi-Ku, Fukuoka 819-0395, Japan

**Keywords:** porous polymer, immobilized catalyst, continuous flow system

## Abstract

Continuous flow reactors with immobilized catalysts are in great demand in various industries, to achieve easy separation, regeneration, and recycling of catalysts from products. Oxidation of alcohols with 4-amino-TEMPO-immobilized monolith catalyst was investigated in batch and continuous flow systems. The polymer monoliths were prepared by polymerization-induced phase separation using styrene derivatives, and 4-amino-TEMPO was immobilized on the polymer monolith with a flow reaction. The prepared 4-amino-TEMPO-immobilized monoliths showed high permeability, due to their high porosity. In batch oxidation, the reaction rate of 4-amino-TEMPO-immobilized monolith varied with stirring. In flow oxidation, the eluent permeated without clogging, and efficient flow oxidation was possible with residence times of 2–8 min. In the recycling test of the flow oxidation reaction, the catalyst could be used at least six times without catalyst deactivation.

## 1. Introduction

In recent years, there has been a great interest in continuous flow technology. Practical and efficient flow reactors have been developed in diverse fields, such as drug discovery, biofunctional substances, petrochemicals, and nanomaterials [1,2,3,4]. Continuous flow reactions are attracting attention as next-generation chemical processes and are being applied industrially. However, some problems remain with the flow system. It is difficult to apply the chemical reaction with catalysts, which have been the most actively studied in organic synthesis. Furthermore, the catalysts elute with the flow of the reaction solution, making it difficult to apply the catalysts to flow processes. Therefore, appropriate immobilized catalysts are required for the flow processes [5,6]. Immobilized catalysts are heterogeneous and have the advantage of easy solid–liquid separation, but they are less reactive than homogeneous catalysts. Immobilized catalysts used in flow reactions must have a large surface area, easy mass transfer, and low-pressure drop-in distribution to successfully take advantage of the characteristics of flow reactions.

There are three main approaches to the immobilized catalysts for continuous flow reactors: wall-coated, particle-packed, and monolithic catalysts [7]. The wall-coated approach minimizes mass transfer resistance and ensures a smooth flow of reagents, but the catalyst loading is lower than the other approaches because the catalyst is supported only on the inner wall of the flow reactor [8]. The particle-packed approach is more versatile because the reactors with catalysts are prepared by simply packing the supports [9]. However, the porosity of the particle-packed column is geometrically limited, resulting in a high-pressure drop for the continuous flow reactor [10]. In contrast, the monolith with three-dimensionally penetrated pores is also utilized as the immobilizing catalyst. The monolith allows both high porosity and small skeleton size in the column, giving excellent target diffusion with low pressure drop. Therefore, the monolithic immobilized catalysts are the best approach for the flow reactors [11,12,13,14].

Both inorganic and organic (polymer) monoliths have been studied. As inorganic monoliths, silica monoliths have been studied intensively [15,16,17]. Polymer monoliths have the advantage that the functional groups and backbones can be designed, allowing the materials to be tailored to the application. Svec and Fréchet developed various polymer monoliths with polymerization-induced phase separation under radical polymerization [18,19]. The Uyama group reported polymer monoliths and applications such as adsorption matrix and separation, using thermally induced phase separation [20,21].

Our group has been investigating polymeric monoliths as supports for immobilized catalysts for flow synthesis [11,12,13,14]. We have reported on the development of monolith immobilization catalysts of metal nanoparticles, organometallic complexes, and organocatalysts. For monolithic immobilized catalysts with metal nanoparticles and organometallic complexes, we have reported the development of immobilized catalyst reactors without catalyst leakage. In the organocatalyst monoliths, we have shown that the excellent mass transfer of the monoliths results in high reactivity.

Our previous work investigated the continuous flow synthesis of cross-coupling and aldol condensation. In the chemical industry, more fundamental chemical reactions, such as oxidation and reduction, are in great demand. In oxidation reactions, several groups reported the fabrication of immobilized 2, 2, 6, 6-tetramethylpiperidine 1-oxyl (TEMPO) catalyst, a typical organocatalyst for oxidation reactions with silica particles, magnetic particles, and fibers [22,23,24]. In this study, we investigated the preparation of polymer monoliths immobilized with 4-amino-2, 2, 6, 6-tetramethylpiperidine 1-oxyl (4-amino-TEMPO), a derivative of TEMPO. The properties of 4-amino-TEMPO-immobilized monolith were investigated with batch and continuous flow reactions. The polymer monolith of poly (4-chloromethyl-styrene-*co*-divinylbenzene) was prepared by polymerization-induced phase separation. Subsequently, 4-amino-TEMPO was immobilized on the polymer by a substitution reaction. 4-Amino-TEMPO-immobilized monolith was investigated given its physical properties and catalytic activity. The porous structure of the monoliths was measured by scanning electron microscope (SEM), mercury intrusion porosimeter, and permeability measurements. The catalytic activity was analyzed by oxidation of alcohols in batch and continuous flow conditions. Under flow conditions, differences in reactivity and oxidation reactions of various substrates were examined, due to differences in residence time. The recyclability of the immobilized catalyst was also examined (Figure 1).

## 2. Experimental Section

### 2.1. Materials

Water with a low efficiency of 18.2 MΩ cm (Milli-Q, Millipore Co., Bedford, MA, USA) was used in all experiments. 4-chloromethyl-styrene (Tokyo Chemical Industry Co., Ltd., Tokyo, Japan) and divinylbenzene (DVB, Sigma-Aldrich Co., St. Louis, MO, USA) were used after removing the stabilizer by passing through the alumina column. Other reagents were used without purification: 4-amino-TEMPO, *N,N*-diisopropylethylamine (DIPEA), tetrabutylammonium bromide (TBAB), tert-butyl alcohol (*^t^*BuOH), 2-methoxybenzyl alcohol, 4-methoxybenzyl alcohol, 4-nitrobenzyl alcohol, (Tokyo Chemical Industry Co., Ltd., Tokyo, Japan), 2,2′-azobis (isobutyronitrile) (AIBN), dichloromethane (CH_2_Cl_2_), *N,N*-dimethylformamide (DMF), (+/−)-1-phenylethanol, 2-phenylethanol, tetrahydrofuran (THF), toluene (FUJIFILM Wako Pure Chemical Co., Osaka, Japan), benzyl alcohol, 1-octanol (Kanto Chemical Co., Tokyo, Japan), and *m*-chloroperoxybenzoic acid (*m*-CPBA, Merck Co., Darmstadt, Germany). A stainless-steel column (0.628 mL, 4 mm i.d., 50 mm length) was used for monolith synthesis.

### 2.2. Characterization

The samples for field emission scanning electron microscope (FE-SEM) analysis were coated with platinum (thickness: approx. 4 nm) using a JEOL JFC-1600 auto fine coater (JEOL Ltd., Tokyo, Japan). FE-SEM analysis was performed with a Hitachi SU8000 microscope (Hitachi High-Technologies Corporation, Tokyo, Japan). High-performance liquid chromatography (HPLC) analyses were performed on a JASCO LC-2000Plus system equipped with a JASCO DG-980-50 degasser, a JASCO PU-980 pump, a Kanto Chemical Mightysil RP-18 GP 250-4.6 column (Kanto Chemical Co., Tokyo, Japan), a JASCO UV-2077Plus UV detector, and a JASCO CO-2065Plus column oven (JASCO Co., Tokyo, Japan). Acetonitrile/water (40:60 or 50:50 *v*/*v*) containing trifluoroacetic acid (0.1 vol%) was employed as a mobile phase in the HPLC measurements. Mercury intrusion porosimetry (MIP) analysis was performed using a Micromeritics AutoPoreIV9520 (Micromeritics Instrument Co., Norcross, GA, USA). Elemental analysis was performed using a Yanaco CHN coder MT-5 (Yanaco technical science Co., Ltd., Tokyo, Japan). ESR measurements were performed using an EMX Model 8/2.7 ESR instrument (Bruker Co., Billerica, Massachusetts, USA). FT-IR measurements were performed using FT/IR-620 (JASCO Co., Tokyo, Japan). A syringe pump (YSP-101, YMC Co., Ltd., Kyoto, Japan) and a pressure gauge (KDM-30, Krone Co., Tokyo, Japan) were used for permeability measurements. The flow behavior analysis was performed using a sample injector (VI-II, FLOM Co., Tokyo, Japan), a Z-type flow cell (FIA-ZSMA-20-TEF, Ocean Optics Inc., Dunedin, FL, USA), a halogen lamp (HL-2000, Ocean Optics Inc.), and a UV-visible spectrometer (USB2000+, Ocean Optics Inc.).

### 2.3. Preparation of Poly (4-Chloromethyl-Styrene-co-DVB) Monolith (Cl−Monolith)

4-Chloromethyl-styrene (0.444, 0.389, 0.333 and 0.278 mL, 100-X wt%), DVB (0.132, 0.198, 0.264 and 0.330 mL, X = 20, 30, 40 and 50 wt%), toluene (0.303 mL), 1-octanol (0.765 mL) and AIBN (1 wt% respective to the total monomers) were added to each vial [25]. Each monomer solution was nitrogen purged for 30 min, injected into stainless-steel columns, and polymerized at 70 °C for 12 h. The prepared poly (4-chloromethyl-styrene-*co*-DVB) monoliths (Cl−Monolith) were washed with THF.

### 2.4. Preparation of 4-Amino-TEMPO-Immobilized Monolith (Monolith)

4-Amino-TEMPO was immobilized on Cl−Monolith prepared using a stainless-steel column (5 cm). The column-polymerized DVB 20, 30, 40, and 50 wt% monoliths were permeated with 4-amino-TEMPO (1 eq. for Cl groups in the monolith), DIPEA (1.2 eq.) and DMF (12 mL) at 0.75 mL/h flow rate, and 60 °C. After the reaction, the columns were washed with Water and THF, and the column solution was replaced with CH_2_Cl_2_/*^t^*BuOH (1:1, *v*/*v*).

### 2.5. Permeabilities of Continuous-Flow Reactor

Cl−Monolith and Monolith made using stainless-steel columns were connected to a syringe pump with a gas-tight syringe connected and a pressure gauge. The CH_2_Cl_2_/*^t^*BuOH (1:1, *v*/*v*) permeabilities of the columns were examined, to obtain permeabilities using Darcy’s law (*k*_D_), as follows:(1)$$\frac{\Delta \mathrm{PA}}{\mu L}=\frac{1}{k_{D}}Q$$
where Δ*P*, μ, *L*, *A*, and *Q* are the pressure loss, viscosity, length, base area, and flow rate of elution, respectively [12].

### 2.6. Flow Behavior Analysis

Flow behavior was analyzed by pulse tracer to determine porosity and internal structure [26]. A syringe pump with a gas-tight syringe, a sample injector, a stainless-steel column packed with Cl-Monolith−1 or Monolith−1, and a Z-type flow cell, were connected. Benzene (M.W. = 7.8 × 10^1^) and polystyrene (M.W. = 1.21 × 10^3^, 5.51 × 10^4^) (2 mg/mL) were injected as tracer molecules, via an injector. The absorbance of the tracer molecules was measured using a halogen lamp and a UV-visible spectrometer. CH_2_Cl_2_/*^t^*BuOH (1:1, *v*/*v*) was used as the solvent, at a flow rate of 9.42 mL/h (residence time 4 min). When the tracer is injected into the column, the volume of the tracer eluted (*V_e_*) can be calculated as follows:(2)$$V_{e}=V_{0}+KV_{p}$$
where *V_p_* and *V*_0_ are the amounts of solvent inside and around the gel (voids), respectively. *K* represents the distribution coefficient of the tracer and ranges from 0 to 1. If high molecular weight tracers, such as proteins, are completely excluded from the inner phase of the gel, *K* is 0. However, low molecular weight ones, such as eluents, can diffuse freely into the gel, so *K* is ~1 [27,28]. The porosity (*ε_g_*) can be estimated using *V_e_*, the geometric volume of the empty column (*V_c_*), the flow rate (v), and the peak absorbance time (t) as follows.
(3)$$\varepsilon_{g}=\frac{V_{e}}{V_{c}}=1-\frac{\left(V_{c}-v\times t\right)}{V_{c}}\;\;$$

*V_e_* was obtained from Equation (2), and porosity was calculated using Equation (3).

### 2.7. Oxidation Reaction with Immobilized Catalyst

Benzyl alcohol (1.5 mmol, 1 eq.) as substrate, *m*-CPBA (1.8 mmol, 1.2 eq.) as co-oxidant, TBAB (0.03 mmol, 0.02 eq.) as co-catalyst, nitrobenzene (0.24 mmol) as internal standard CH_2_Cl_2_/*^t^*BuOH (1:1, *v*/*v*, 10 mL), 4-amino-TEMPO and finely ground Monolith−1 (0.075 mmol, 0.05 eq.), were added to the solution for oxidation reaction at room temperature with stirring (200 rpm), and without stirring. The reaction solution was diluted 100-fold with MeCN, and the product yield was determined by HPLC (MeCN: water = 40:60, *v*/*v*).

The oxidation of alcohols was also examined by continuous flow reactions [24]. Benzyl alcohol (0.15 mol/L, 1 eq.), *m*-CPBA (0.18 mol/L, 1.2 eq.), TBAB (0.003 mol/L, 0.02 eq.), nitrobenzene (0.024 mol/L), and CH_2_Cl_2_/*^t^*BuOH (1:1, *v*/*v*) were mixed, and then the mixture was permeated through a Monolith−1 at room temperature using a gas-tight syringe at residence times of 2, 4, and 8 min (18.84, 9.42, and 4.71 mL/h). The reaction substrates were changed from benzyl alcohol to 2-phenylethanol, (+/−)-1-phenylethanol, 4-nitrobenzyl alcohol, 2-methoxybenzyl alcohol, or 4-methoxybenzyl alcohol, and the same experiments were conducted with a residence time of 4 and 8 min. Monolith−1 column was washed with CH_2_Cl_2_/*^t^*BuOH (3.5 mL) following the reaction with benzyl alcohol at a residence time of 4 min, and the benzyl alcohol oxidation reaction was repeated at a residence time of 4 min. The permeate was collected, diluted 100-fold with MeCN, and run by HPLC (MeCN: water = 40:60 or 50:50 *v*/*v*).

## 3. Results and Discussion

### 3.1. Preparation of Monolith (4-Amino-TEMPO-Immobilized Monolith)

Cl−Monoliths were prepared by radical polymerization using a porogen of 1-octanol, and the obtained Cl−Monoliths were modified with 4-amino-TEMPO (Figure 2). The structure was optimized by changing the DVB content ratio to 20, 30, 40, and 50 wt% (Percentage of total monomers).

From the SEM images in Figure 3 and Appendix A, it was observed that all the Cl−Monoliths and Monoliths had a porous structure with interconnected particles and macro-pores between the particles. The particle size of the monoliths with higher DVB content was smaller, and the shape of the monoliths was maintained after immobilization with 4-amino-TEMPO (Table 1 and Appendix A). The pore diameters after 4-amino-TEMPO immobilization were as follows: around 0.5–70 µm for Monolith−1, around 1–30 µm for Monolith−2, around 1–20 µm for Monolith−3, and 0.1–2 µm for Monolith−4, respectively (Figure 3 and Figure 4). A slight change in pore size was also observed by the 4-amino-TEMPO immobilization reaction (Figure 4 and Appendix A), and the porosity of each monolith was almost the same (65, 61, 59, and 64%, respectively). The amount of 4-amino-TEMPO immobilization was in the order of Monolith−1 (Appendix A), Monolith−2, Monolith−3, and Monolith−4 (Table 1), because smaller DVB content results in a large amount of 4-chloromethyl styrene for 4-amino-TEMPO immobilization. ESR and FT-IR data also suggested that 4-amino-TEMPO was immobilized in Monolith−1 (Appendix A). The following experiments were conducted using Monolith−1 (DVB 20 wt%).

Permeability of the monoliths was in the order of Monolith−1 > Monolith−2 >> Monolith−3, and −4 based on the pore diameters (Table 1), meaning higher permeabilities were observed in the monoliths with lower DVB content. There was no significant difference between before and after the immobilization of 4-amino-TEMPO to monoliths.

The porosity of the monolith in the swollen state was measured by tracers of different sizes and was changed with the size of the tracers (Appendix A). When benzene was used as a tracer, the porosity was around 75% displaying higher porosity than that from MIP. The swelling of the monoliths by the solvent suggests gel properties and benzene diffusion into the monoliths’ interior [13]. The high porosity in the swelling state suggests that Monolith−1 provides the catalyst with the efficient mass transfer of the substrate.

### 3.2. Batch Oxidation Reaction

Homogeneous and heterogeneous batch oxidation of benzyl alcohol was investigated, where the homogeneous catalyst used was 4-amino-TEMPO and the heterogeneous catalyst used was Monolith−1. Under stirred conditions, the homogeneous (4-amino-TEMPO) catalyst reached near-equilibrium after 6 min, yielding 59% and TON 12 benzaldehyde, with a benzaldehyde selectivity of 96%. Whereas the heterogeneous monolithic catalyst (Monolith−1) reached equilibrium after 10 min, yielding 60% and TON 12 benzaldehyde, with a benzaldehyde selectivity of 94% (Figure 5a). Without stirring, the reaction almost reached equilibrium after 6 min with the homogeneous catalyst, yielding 59% and TON 12 benzaldehyde, with a benzaldehyde selectivity of 92% which is similar to the reaction under a stirred condition. On the other hand, the reaction reached equilibrium after 12 min with the heterogeneous catalyst (Monolith−1), yielding 64% and TON 13 benzaldehyde, with a benzaldehyde selectivity of 93% (Figure 5b). A comparison of the TON using Monolith−1 after 6 min, showed 11 for the condition with stirring, and 9.3 for the condition without stirring. Surprisingly, under a stirred condition, the reaction rate of the heterogeneous monolithic catalyst approached that of the homogeneous catalyst. Although benzoic acid was formed as a side reaction within each reaction, the selectivity of benzaldehyde synthesis was high, ranging from 92 to 96%.

The oxidation of benzyl alcohol to benzaldehyde using Monolith−1 was relatively fast, especially in the stirred system. In many cases, heterogeneous catalyzed reactions are disadvantageous in terms of mass transfer. Our heterogeneous monolithic catalyst was shown to be useful in the reaction because it showed reaction rates approaching that of a homogeneous catalyst. For this heterogeneous monolithic catalyst, stirring induced a difference in the reaction rate.

### 3.3. Flow Oxidation Reaction

The oxidation reaction of benzyl alcohol was examined using Monolith−1 in a continuous flow reaction. When the reaction was carried out by changing the residence time to 2, 4, and 8 min, the yields of benzaldehyde were 46, 48, and 52%, respectively, and the selectivity of benzaldehyde was 93, 94, or 81% (Figure 6a, Appendix A). These results indicate that Monolith−1 allows a fast and selective oxidation reaction to proceed within a short residence time, but with a residence time over 8 min, a by-product of benzoic acid was increased. The yield of the product increased up to 4-column volume but decreased above 6-column volume. When Monolith−1 was washed with solvent after the reaction, benzaldehyde was detected, suggesting a deactivation by the product adsorption.

Monolith−1 was regenerated by washing with solvent after the reaction. After the Monolith−1 was reacted once, it was washed with solvent and utilized again in the reaction (Figure 6b). Interestingly, Monolith−1 showed catalytic activity without deactivation after six uses. Six flow oxidation reactions of benzyl alcohol, including five recycling trials, achieved a TON of 25 and a total benzaldehyde yield of 0.32 g. In addition, the yield at regeneration was higher than the original yield. This may be due to the possibility that benzyl alcohol flowed out in some parts of Monolith−1 during the initial oxidation reaction before the co-oxidant *m*-CPBA diffused out, and the reaction did not proceed. It was considered that the *m*-CPBA diffused reliably into Monolith−1 with each recycling cycle, and the yield was higher because the oxidation reaction could be performed with more immobilized 4-amino-TEMPO activated. The catalyst activity of Monolith−1 was reduced after the flow of 6-column volume, but Monolith−1 was almost quantitatively activated by solvent washing. Monolith−1 can be used many times, and the total catalyst turnover becomes higher because of the reactivation property, suggesting a high industrial advantage.

To confirm the applicability of Monolith−1, the various alcohols were studied (Table 2). The yields of aldehyde or ketone of 2-phenylethanol, (+/−)-1-phenylethanol, 4-nitrobenzyl alcohol, 2-methoxybenzyl alcohol, and 4-methoxybenzyl alcohol are 18, 22, 35, 59, and 62%, respectively, at a residence time of 4 min. The lower yield of oxidation reaction reflected the reactivity of substances. In the case of 2-phenylethanol, the benzyl position of 2-phenylethanol was not oxidized, and phenylacetaldehyde was obtained. The yield was lower because the alcohol was not benzyl-positioned. The yield increased when the residence time was increased to 8 min (Appendix A). (+/−)-1-Phenylethanol is a secondary alcohol and the yield decreased due to steric hindrance. The yield was also improved by increasing the residence time (Appendix A). The reactivity of benzyl alcohol with substituted functional groups was different, and benzyl alcohol with methoxy groups gave higher yields. Although the reactivity of the catalyst for these substrates was different, these results suggest that the flow reaction of 4-amino-TEMPO-immobilized catalysts can be applied to various alcohols.

## 4. Conclusions

4-amino-TEMPO-immobilized monolith was easily prepared by radical polymerization, under the phase separation condition of poly (4-chloromethyl styrene-*co*-DVB) and the subsequent substitution reaction. In the batch reaction, the reactivity of the 4-amino-TEMPO-immobilized monolith catalyst showed high reactivity with stirring. In the continuous flow reaction, the 4-amino-TEMPO-immobilized monolith exhibited a high reactivity and benzaldehyde selectivity, up to 6-column volume based on the efficient mass transfer. The activity of the monolithic catalyst was reduced by the inaccessibility of the substrate to the active point of the catalyst, due to product adsorption to the catalyst. However, washing by solvent flow resulted in regained activity, and a total of six catalytic reactions were achieved, including five recycling tests. The 4-amino-TEMPO-immobilized catalyst was applied to the oxidation reaction of several kinds of alcohols. The catalyst-immobilized porous monolith showed high reactivity in both batch and flow system reactions, suggesting that it is suitable as a support for organocatalysts.

## Figures and Tables

**Figure 1 polymers-14-05123-f001:**
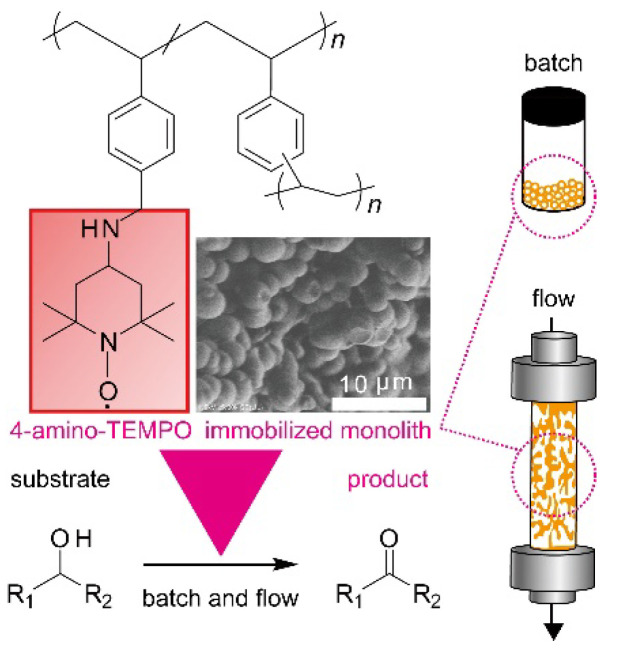
Oxidation of alcohol using 4-amino-TEMPO-immobilized monolith catalyst.

**Figure 2 polymers-14-05123-f002:**
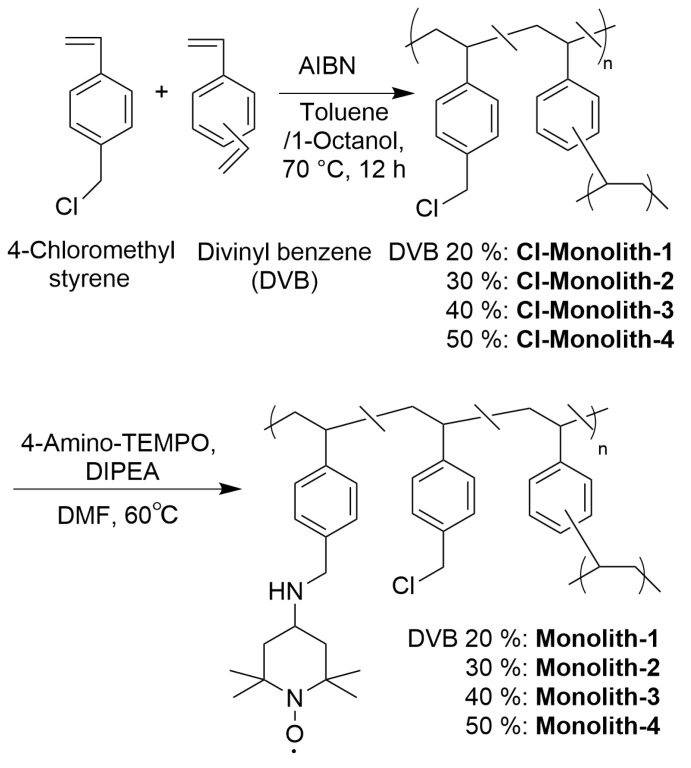
Preparation of Monolith (4-amino-TEMPO-immobilized monolith) with Cl−Monolith.

**Figure 3 polymers-14-05123-f003:**
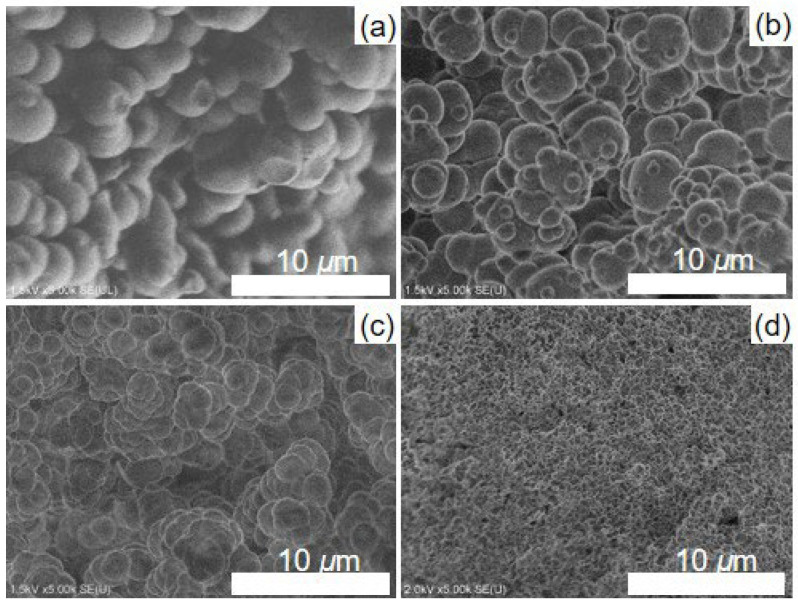
SEM images of Monolith (**a**) −1, (**b**) −2, (**c**) −3, and (**d**) −4.

**Figure 4 polymers-14-05123-f004:**
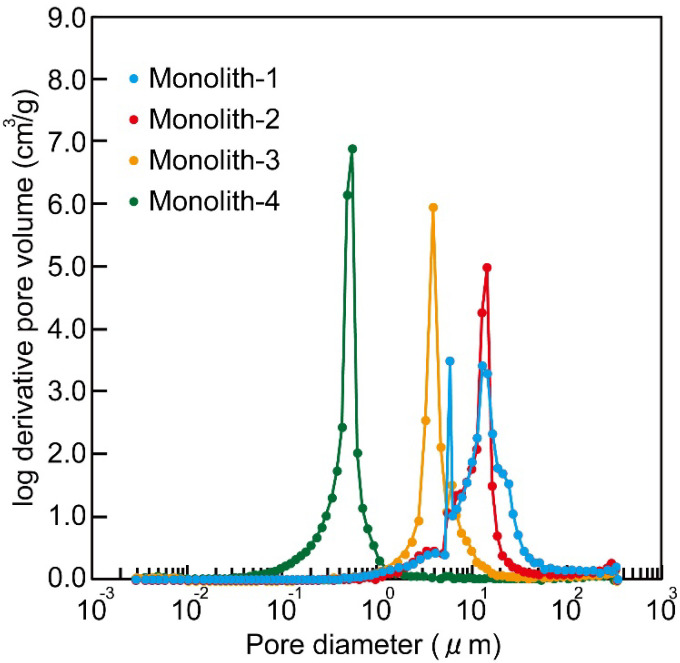
Pore size distribution of Monolith−1, −2, −3, and −4 as characterized by MIP.

**Figure 5 polymers-14-05123-f005:**
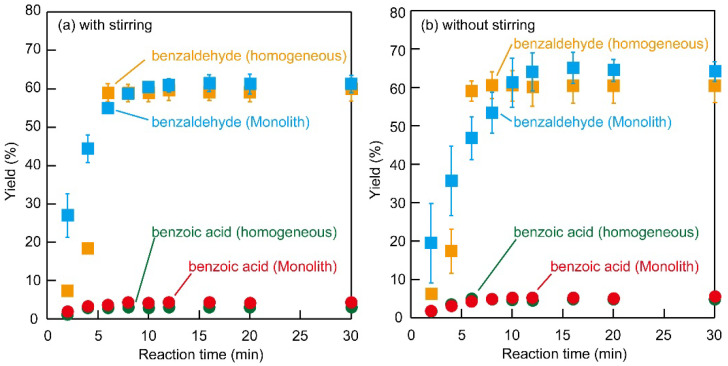
Batch oxidation of benzyl alcohol (**a**) with stirring, and (**b**) without stirring. The yield and reaction conditions are shown as follows (■: benzaldehyde with homogeneous reaction, ■: benzaldehyde with Monolith−1, ●: benzoic acid with homogenous reaction, and ●: benzoic acid with Monolith−1).

**Figure 6 polymers-14-05123-f006:**
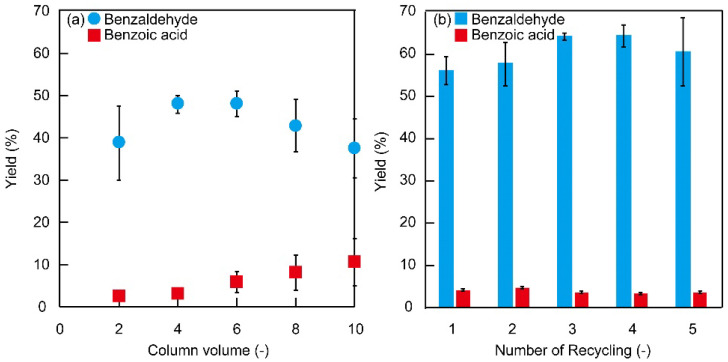
(**a**) Flow oxidation of benzyl alcohol by Monolith−1 at a residence time of 4 min. (**b**) Recycling test for flow oxidation of benzyl alcohol by Monolith−1 at a residence time of 4 min, and the yield of 4-column volume is shown.

**Table 1 polymers-14-05123-t001:** Properties of Monolith−1, −2, −3, and −4.

Material	DVB [wt %]	Specific Surface Area [m^2^ g^−1^] ^a^	Average Pore Diameter [μm] ^a^	Porosity [%] ^a^	4-Amino-TEMPO [mmol/g] ^b^	Permeability [m^2^] ^c^
Monolith−1	20	0.85	8.8	65	0.53	5.7 × 10^−12^
Monolith−2	30	0.74	8.4	61	0.36	4.1 × 10^−13^
Monolith−3	40	18	0.34	59	0.33	7.3 × 10^−14^
Monolith−4	50	29	0.23	64	0.24	3.5 × 10^−14^

^a^ The values were calculated by MIP. ^b^ 4-amino-TEMPO amount is estimated by nitrogen amount of elemental analysis. ^c^ Permeability is based on Darcy’s law.

**Table 2 polymers-14-05123-t002:** Oxidation of various alcohols to the aldehyde and ketone using Monolith−1 after 4-column volume flow. The yield of aldehyde and ketone with a residence time of 4 and 8 min.

Substrate	Product	The Yield of 4 min [%]	The Yield of 8 min [%]
Benzyl alcohol 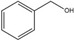	Benzaldehyde 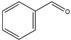	48	52
2-Phenylethanol 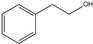	Phenylacetaldehyde 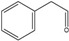	18	48
(+/−)-1-Phenylethanol 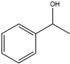	Acetophenone 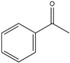	22	39
4-Nitrobenzyl alcohol 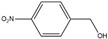	4-Nitrobenzaldehyde 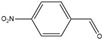	35	30
2-Methoxybenzyl alcohol 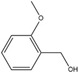	2-Methoxybenzaldehyde 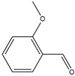	59	59
4-Methoxybenzyl alcohol 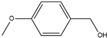	4-Methoxybenzaldehyde 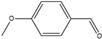	62	72

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
