# Peer review of "4-Amino-TEMPO-Immobilized Polymer Monolith: Preparations, and Recycling Performance of Catalyst for Alcohol Oxidation"

_polymers, 2022, doi:10.3390/polym14235123_

Round 1

Reviewer 1 Report

Dear Editor,

    The present manuscript provided some interesting findings on the continuous flow reaction of alcohol oxidation; in general, the topic is hot and valuable. Unfortunately, I believed that the experimental design is relatively superficial so that the critical information is missed. For instance, the author did not show the innovation of this work in the Introduction, and discussions related to the catalytic mechanism are still confused. More importantly, some of the strange results are exhibited without explanation, such as the sudden increase of yield in Fig. 5. As a result, I suggested a “Reject” for the present manuscript, and I hope the above comments can help for the further submission.

Author Response

We appreciate for your valuable comment. This is our first report on immobilized TEMPO. There are still many unknowns and the discussion might have been inadequate. We have corrected many of these based on your comments. The introduction has been modified significantly to describe the purpose of this study and the importance of immobilized catalysts in the flow synthesis method; for TEMPO immobilized catalysts, we have described the activity and TON of the catalysts and added a discussion.

Reviewer 2 Report

Selection of appropriate immobilized catalysts is essential for continuous flow technology related to production of new drugs, functional substances, petrochemicals and nanomaterials. The feasibility and efficiency of applying TEMPO immobilized monolith catalyst in continuous flow reaction process have been investigated in this submission. The original outcomes presented in this manuscript were well prepared and would be helpful to enhance application of TEMPO immobilized monolith catalyst in continuous flow reactors. It is suggested to accept the submission for publication after further modification, and the followed comments are advised for improving the current manuscript.

(1) This submission is a research article based on the experimental investigation carried out by the authors. It is better to modify the manuscript title for matching the original work. The current title likes that for a review paper.

(2) The basic demands for the immobilized monolith catalyst of alcohol oxidation in the batch reaction and in the continuous flow reaction should make clear in the “Introduction” section of the manuscript.

(3) Please check and modify the editing format of Figures and Tables according to the Guide for Authors of MDPI Journals. For example, Scheme 1(on Page 5)should illustrated as a Figure,  and the subfigure number also should be unified (A or a)

 (4) How exactly to evaluate the reactivity of the immobilized monolith catalysts? It was only by reuse times?

 (5) Please check presentation of several sentences in Abstract and Conclusion sections, such as In the batch system, the reaction rate with (Line 16)”, In flow oxidation, the eluent could (Line 17)”, “Though the monolith catalyst was deactivated due to the product adsorption (Line 302)”,  and “Porous polymer were suitable supports for (Line 305)”.

Author Response

We appreciate for your valuable comments. We have incorporated your comments and made significant revisions to the paper. The followings are the summary of our responses to your questions. We hope that the manuscript is accepted and published in “Polymers”

Q1. This submission is a research article based on the experimental investigation carried out by the authors. It is better to modify the manuscript title for matching the original work. The current title likes that for a review paper.

A1.  P1, line 2, We appreciate for the comment. The title was corrected as suggested. New title is “4-Amino-TEMPO immobilized polymer monolith: preparations, and recycling performance of catalyst for alcohol oxidation.”

Q2. The basic demands for the immobilized monolith catalyst of alcohol oxidation in the batch reaction and in the continuous flow reaction should make clear in the “Introduction” section of the manuscript.

Ans 2, P2, Introduction We appreciate for the comment. The “Introduction” was significantly corrected as suggested. The basic demand and the principle of the monolith immobilization catalyst were added in the introduction part as follows. 

The basic demands for the immobilized monolith catalyst of alcohol oxidation are “The monoliths allow both high porosity and small skeleton size in the column, giving excellent target diffusion with low-pressure drop. (line )”, “In the chemical industry, more fundamental chemical reactions like oxidation and reduction are of great demand. (line)”

Q3.  Please check and modify the editing format of Figures and Tables according to the Guide for Authors of MDPI Journals. For example, Scheme 1(on Page 5)should illustrated as a Figure,  and the subfigure number also should be unified (A or a)

Ans 3, Figures and Tables,  We appreciate for the comment. The editing format of Figures and Tables was corrected as suggested.

Q4. How exactly to evaluate the reactivity of the immobilized monolith catalysts? It was only by reuse times?

Ans.4, P9, line 5, We appreciate for the comment. As you pointed out the most important point in this catalyst is the reuse. But of course, it is not enough to evaluate the catalyst as you suggested. We evaluate the catalyst the reaction selectivity and total TON. In many reports of TEMPO immobilized catalyst, most of the researchers described the yield of aldehyde, but aldehyde and carboxylic acids should be yielded. The selectivity is high in the flow reaction. We also added the TON of the immobilized catalyst, and the amount of product in the Result and Discussion part.

Q5. Please check presentation of several sentences in Abstract and Conclusion sections, such as “In the batch system, the reaction rate with (Line 16)”, “In flow oxidation, the eluent could (Line 17)”, “Though the monolith catalyst was deactivated due to the product adsorption (Line 302)”,  and “Porous polymer were suitable supports for (Line 305)”.

Ans 5. P2, and 10, We appreciate for the comment. The English is carefully corrected. The English of Abstract and Conclusion was corrected as suggested.

Reviewer 3 Report

This work described the synthesis of TEMPO immobilized monoliths and their catalytic applications in oxidation of alcohols. The polymeric monoliths of poly(4-chloromethyl-styrene-co-divinylbenzene) were prepared by the polymerization induced phase separation, followed by the substitution reaction with 4-amino-TEMPO for TEMPO immobilization. The polymeric monoliths were characterized by SEM, Mercury intrusion porosimetry, elemental analysis. Oxidation of alcohols with TEMPO immobilized monolith catalyst was investigated in batch and continuous flow systems. The TEMPO-immobilized monoliths showed high permeability due to their high porosity, great reactivity, and recyclability up to 6 times without catalyst deactivation. I would like to see an updated manuscript before a decision of acceptance since I am not totally convinced that this work is suitable for Polymers. 

1. I think the introduction is lacking of motivation and background. What is the state-of-art polymeric monolith for flow reaction? Why do you need to develop such polymeric monoliths? Why do you want to use such monoliths in an oxidation reaction? What impact do you envision your new monoliths bring to the community? After reading the introduction, I am not totally convinced that this work is crucial to polymer peers or catalysis peers and will be impactful.  

2. In figure, what are the yellow and blue color stand for?

3. Please perform ERP for providing more accurate measurement on TEMPO loading.

4. In table 1, why the porosity of monolith 4 is higher than 1, 2, and 3?

5. Please unify the format of front of numbers in table 1.

6. Please provide error bar for Figure 4 and 5.

7. In figure 5, why the yield of 3rd and 4th regeneration was higher than the original yield?

8. Table 2 is confusing. Please provide a better table 2. 

9. I would like to see a gram scale reaction since the reaction scale now (~100 mg) is too small. 

Author Response

We appreciate for your valuable comments. We have incorporated your comments and made significant revisions to the paper. The followings are the summary of our responses to your questions. We hope that the manuscript is accepted and published in “Polymers”

Q1.

I think the introduction is lacking of motivation and background. What is the state-of-art polymeric monolith for flow reaction? Why do you need to develop such polymeric monoliths? Why do you want to use such monoliths in an oxidation reaction? What impact do you envision your new monoliths bring to the community? After reading the introduction, I am not totally convinced that this work is crucial to polymer peers or catalysis peers and will be impactful.

Ans. 1

P2. Introduction,

We appreciate for the comment.  We significantly corrected the Introduction part as suggested. We described the importance of immobilized catalyst, specially in the flow condition. We added the advantage of polymer monolith with other catalyst supports. We also added the motivation of this work.  

Q2.

In figure, what are the yellow and blue color stand for?

Ans. 2

P3, Figure 1,

We appreciate for the comment. The color was corrected as orange (polymer ) and white (hollow). The color was changed for easy understanding.

Q3.

Please perform EPR for providing more accurate measurement on TEMPO loading.

Ans. 3

Supporting Information,

We appreciate for the comment.  EPR spectra were measured in order to evaluate the amount of TEMPO. Though we could observe the EPR spectra of TEMPO-immobilized monolith and TEMPO. We tried to calibrate the peak but it was not successful, which might be due to the high density of TEMPO in the monolith. It is still under investigation.

We also measured FTIR spectra of Monolith-1 to confirm the immobilization of TEMPO.

Q4.

In table 1, why the porosity of monolith 4 is higher than 1, 2, and 3?

Ans. 4

P7, Table 1, We appreciate for the comment. The porosity is a little different with each other. But we did not regard the porosity difference as significant (59- 65 %). The monolith formation is based on the polymerization induced phase separation and DVB affected the phase separation.

Q5.

Please unify the format of front of numbers in table 1.

Ans. 5

P7, Table 1, We appreciate for the comment. The font format was corrected as suggested.

Q6.

Please provide error bar for Figure 4 and 5.

Ans. 6

P8, and P9, Figure 5 and 6 (in the original version Figure 4 and 5).

We appreciate for the comment. The error bars were added in Figure 5 and Figure 6.

Q7.

In figure 5, why the yield of 3rd and 4th regeneration was higher than original yield?

Ans. 7

P9, Figure 6, line 6

We appreciate for the comment. The discussion of the higher activity was added in the discussion part as follows.

  “This may be due to the possibility that benzyl alcohol flowed out in some parts of Monolith-1 during the initial oxidation reaction before the co-oxidant m-CPBA diffused out and the reaction did not proceed. It was considered that the m-CPBA diffused reliably into Monolith-1 with each recycling cycle, and the yield was higher because the oxidation reaction could be performed with more immobilized 4-amino-TEMPO activated.”

Q8.

Table 2 is confusing, please provide a better table 2.

Ans 8

P10, Table 2

We appreciate for the comment. Table 2 was corrected as suggested.

Q9.

I would like to see a gram scale reaction since the reaction scale now (~100 mg) is too small.

Ans 9

P9, line 5,

We appreciate for the comment. The practical yield was added in Result and Discussion. The yield of the 5 cycle was 0.32g. 

Round 2

Reviewer 1 Report

Dear Editor,

    The author make a great job in the revision stage, and the logical relationship is easier to follow. There is one last suggestion in Table 2, where the substrates and products can be exhibited in the forms of structural formula. After that, the present manuscript is suggested to be accepted.

Author Response

Reviewer 1

The author make a great job in the revision stage, and the logical relationship is easier to follow. There is one last suggestion in Table 2, where the substrates and products can be exhibited in the form of structural formula. After that, the present manuscript is to be accepted.

Ans.

We greatly appreciate your advice on logical explanations. We also appreciate the comment about Table 2. Table 2 was corrected as suggested. The structural formulae of the substrates and products are exhibited in new Table 2.

Reviewer 3 Report

I would like to thank the authors making efforts to address my concerns. I am happy with the revised manuscript and the response, expect the question #3 about EPR. EPR spectra in the SI was great and I would recommend the authors establishing a calibration curve for calculation. First, a series of standard solutions of TEMPO free radical with different concentrations are prepared and their EPR spectra are measured. Then, the EPR signal intensities were determined by double-integration and plotted against the concentration to generate a standard concentration curve. Then TEMPO-immobilized monolith can be measured in the same way and was ascertained by using the standard concentration curve.

Author Response

We appreciate for the comment. We involved your comment as follows.

Reviewer3

I would like to thank the authors make efforts to address my concerns. I am happy with the revised manuscript and the response, except the question #3 about EPR. EPR spectra in the SI were great and I would recommend the authors establish a calibration curve for calculation. First, a series of standard solutions of TEMPO free radical with different concentrations are prepared and their EPR spectra are measured. Then, the EPR signal intensities were determined by double-integration, and plotted against the concentration to generate a standard concentration curve. Then TEMPO-immobilized monolith can be measured in the same way and was ascertained by using the standard concentration curve.

Ans.

We greatly appreciate your advice on EPR spectra. Figure S4 was corrected as suggested. For the EPR measurement, Monolith-1 was insoluble in the solvent, and the peak shape did not match that of the TEMPO solution even when dispersed in the solvent. Therefore, measurements were made using solid TEMPO and solid Monolith-1. ESR spectra of three different amounts of TEMPO were measured and double integrated to produce a calibration curve. The ESR spectra were then measured using three different amounts of Monolith-1 and double integrated to estimate the amount of 4-amino-TEMPO immobilized. (Figure S4)  Because the measurements were performed in solid form, we are concerned that the rate of radical interactions was not constant, and that quantitative discussions were considered to contain uncertainties.